# Construction and validation of a gene expression classifier to predict immunotherapy response in primary triple-negative breast cancer

Miquel Ensenyat-Mendez[1], Javier I. J. Orozco [2], Pere Llinàs-Arias[1], Sandra Íñiguez-Muñoz[1], Jennifer L. Baker[3], Matthew P. Salomon[4], Mercè Martí[5,6], Maggie L. DiNome [7], Javier Cortés[8,9,10] & Diego M. Marzese [1✉]

**Abstract**

**Background** Immune checkpoint inhibitors (ICI) improve clinical outcomes in triple-negative breast cancer (TNBC) patients. However, a subset of patients does not respond to treatment. Biomarkers that show ICI predictive potential in other solid tumors, such as levels of PD-L1 and the tumor mutational burden, among others, show a modest predictive performance in patients with TNBC.

**Methods** We built machine learning models based on pre-ICI treatment gene expression profiles to construct gene expression classifiers to identify primary TNBC ICI-responder patients. This study involved 188 ICI-naïve and 721 specimens treated with ICI plus chemotherapy, including TNBC tumors, HR+/HER2− breast tumors, and other solid non-breast tumors.

**Results** The 37-gene TNBC ICI predictive (TNBC-ICI) classifier performs well in predicting pathological complete response (pCR) to ICI plus chemotherapy on an independent TNBC validation cohort (AUC = 0.86). The TNBC-ICI classifier shows better performance than other molecular signatures, including PD-1 (*PDCD1*) and PD-L1 (*CD274*) gene expression (AUC = 0.67). Integrating TNBC-ICI with molecular signatures does not improve the efficiency of the classifier (AUC = 0.75). TNBC-ICI displays a modest accuracy in predicting ICI response in two different cohorts of patients with HR + /HER2- breast cancer (AUC = 0.72 to pembrolizumab and AUC = 0.75 to durvalumab). Evaluation of six cohorts of patients with non-breast solid tumors treated with ICI plus chemotherapy shows overall poor performance (median AUC = 0.67).

**Conclusion** TNBC-ICI predicts pCR to ICI plus chemotherapy in patients with primary TNBC. The study provides a guide to implementing the TNBC-ICI classifier in clinical studies. Further validations will consolidate a novel predictive panel to improve the treatment decision-making for patients with TNBC.

**Plain language summary**

Triple-Negative Breast Cancer (TNBC) is an aggressive type of breast cancer, responsible for a substantial burden of breast cancer-related deaths. In recent years, immunotherapy, a therapy that triggers the patient's immune system to attack the tumor, has arisen as a promising treatment in various cancers, including TNBC. However, a subset of patients with TNBC does not respond to this treatment. Here, we employed advanced computational techniques to predict response to immunotherapy plus chemotherapy in patients with primary TNBC. Our method is more accurate than using other existing markers, such as PD-L1, but is not very accurate in patients with non-TNBC breast cancers or non-breast cancers. This method could potentially be used to better select patients for immunotherapy, upfront, avoiding the side effects and costs of treating patients in which immunotherapy might not work.

---

[1] Cancer Epigenetics Laboratory at the Cancer Cell Biology Group, Health Research Institute of the Balearic Islands (IdISBa), Palma, Spain. [2] Saint John's Cancer Institute, Providence Saint John's Health Center, Santa Monica, CA, USA. [3] Department of Surgery, David Geffen School of Medicine at UCLA, Los Angeles, CA, USA. [4] Department of Medicine, University of Southern California (USC), Los Angeles, CA, USA. [5] Immunology Unit, Department of Cell Biology, Physiology, and Immunology, Institut de Biotecnologia I Biomedicina (IBB), Universitat Autònoma de Barcelona (UAB), Bellaterra, Barcelona, Spain. [6] Biosensing and Bioanalysis Group, Institute of Biotechnology and Biomedicine, Universitat Autònoma de Barcelona, Bellaterra, Barcelona, Spain. [7] Department of Surgery, Duke University School of Medicine, Durham, NC, USA. [8] International Breast Cancer Center (IBCC), Pangaea Oncology, Quironsalud Group, Barcelona, Spain. [9] Medical Scientia Innovation Research (MedSIR), Barcelona, Spain. [10] Faculty of Biomedical and Health Sciences, Department of Medicine, Universidad Europea de Madrid, Madrid, Spain. ✉email: Diego.Marzese@ssib.es

Triple-Negative Breast Cancer (TNBC) is a highly hetero-geneous disease defined by the absence of estrogen receptor (ER), progesterone receptor (PR), and the lack of over-expression of human epidermal growth factor receptor 2 (HER2)[1–3]. Due to the absence of effective therapeutic targets, chemotherapy has been the primary systemic treatment for early and advanced TNBC for decades[4]. This scenario has encouraged the search for new therapeutic agents, such as PARP inhibitors for patients with germline mutations on the *BRCA1/2* genes and, more recently, immune checkpoint inhibitors (ICI)[5,6]. Implementation of ICI has shown significant improvements in the survival and clinical management of patients with different solid tumors. The phase 3 clinical trial IMpassion130 showed that the combination of nab-paclitaxel with atezolizumab — a humanized antibody that restores the antitumoral immune response blocking PD-L1 expressed by tumoral cells— increases the progression-free survival (PFS) of patients with unresectable locally advanced or metastatic TNBC (mTNBC), with a greater improvement noted in patients with PD-L1-positive tumors[7]. While atezolizumab failed to demonstrate a statistically significant improvement in overall survival (OS) in the complete cohort, it showed a clinically meaningful improvement in OS in the PD-L1-positive patients[8,9]. On the other hand, the phase 3 clinical trial KEY-NOTE-355, demonstrated that adding pembrolizumab — a humanized anti-PD-1 antibody that activates exhausted PD-1+ T-cells by preventing interaction with PD-L1— to chemotherapy (including paclitaxel; nab-paclitaxel, or gemcitabine plus carbo-platin) improves PFS and OS in patients with PD-L1-positive untreated, locally recurrent, inoperable, or mTNBC[10]. Thus, solid evidence shows that beneficial results are limited to a subset of patients with high PD-L1 levels in mTNBC[11].

Recent studies have shown the efficacy of this approach in primary TNBC[12,13]. Consequently, in 2021, following the results from KEYNOTE-522[14,15], the FDA approved the use of pem-brolizumab in combination with chemotherapy as neoadjuvant treatment for high-risk primary TNBC. Unlike in the metastatic setting, the addition of pembrolizumab to current treatments resulted in higher rates of pathological complete response (pCR) regardless of PD-L1 levels, although patients with PD-L1-positive tumors did show higher rates (68.9%) than those with PD-L1-negative tumors (45.3%)[15]. Nevertheless, a significant number of patients still do not achieve a pCR in the primary TNBC setting[16,17]. This evidence highlights the need for precision bio-markers that can identify patients with primary TNBC who will benefit from the addition of ICI to chemotherapy.

Beyond PD-L1, other biomarkers that have shown predictive potential in other solid tumors[18,19] may partially explain the benefit of ICI in patients with TNBC. The presumption that '*more genomic alterations generate more neo-antigens*' supports the idea that the higher the tumor mutational burden (TMB), the better the response to ICI treatment. However, recent clinical trials show a modest predictive potential of TMB for ICI response in patients with TNBC, and controversy over the methods to test TMB and PD-L1 and the optimal cutoffs for these variables still remains[20,21].

The development and adaptation of machine learning methods, such as deep learning, allow for the extraction of informative features within the layers of a neural network, and deep learners, among other machine learning methods, allow for the integration of diverse data sources from imaging, clinical covariates, histology, and mole-cular profiling. These agnostic approaches generate robust classifiers from diverse and complex data types, such as imaging, clinical, histology, and molecular profiling[22,23]. Here, we employed machine learning to construct and evaluate gene expression-based signatures that efficiently predict pCR to ICI plus chemotherapy in patients with primary TNBC treated in the phase II/III I-SPY2 clinical trial[24,25]. Thus, we constructed and tested a TNBC ICI response

(TNBC-ICI) predictive classifier that involves 37 genes. The efficacy of TNBC-ICI was compared against other recognized biomarkers and molecular features that have shown modest to good ICI pre-dictive performance. TNBC-ICI has a significant performance in identifying patients with TNBC that are likely to reach pCR to the treatment combining ICI and chemotherapy. This classifier was further tested in ICI-treated cohorts including hormone receptor (HR)-positive/HER2-negative breast cancer (BC) patients, other immune hot and cold non-BC solid tumors (bladder, esophageal, melanoma, and renal), and in an ICI-naïve cohort of patients with TNBC treated with chemotherapy. Here, we provide the details to replicate the TNBC-ICI classifier and a comparison of this resource with other gene expression signatures, molecular features, and clas-sifiers that can predict response to ICI. The validation of this clas-sifier in additional ICI-treated cohorts opens an avenue to improve the management of patients with primary TNBC by identifying upfront cases that are likely going to reach pCR.

## Methods

**Patient selection and data inclusion**. Clinical and gene expres-sion data from a cohort of 759 pretreatment specimens from breast cancer stage II/III patients included in the phase II I-SPY2 clinical trial[24,25] were obtained from the Gene Expression Omnibus (GEO; GSE173839 and GSE194040). We selected gene expression profiling from 50 patients with confirmed TNBC and 90 patients with HR+/HER2– disease before neoadjuvant therapy with durvalumab (anti-PD-L1 antibody) combined with olaparib (PARP inhibitor) and paclitaxel, or with pembrolizumab (anti-PD-1 antibody) combined with paclitaxel. An additional cohort of 56 patients with TNBC treated with paclitaxel from the I-SPY2 cohort was employed to validate the accuracy of the classifier. During the classifier construction, patients who achieved a pCR (ypT0/is, ypN0)[26], defined as the absence of invasive cancer in the breast and regional nodes at the time of surgery, were classified as 'responders'; whereas those who had residual disease were con-sidered 'non-responders'. An additional ICI-naïve TNBC tumor cohort (n = 132) from The Cancer Genome Atlas (TCGA) of chemotherapy-treated patients was obtained from the National Cancer Institute Genomic Data Commons in June 2020 using R/TCGABiolinks (v.2.16.4). Clinical annotations were manually curated using the scanned clinical reports according to ASCO/CAP guidelines to solve issues for samples with missing or dis-crepant clinical or pathological data. Furthermore, gene expres-sion profiles from 581 non-BC ICI-treated solid tumors were obtained from different clinical studies. A complete list of the clinical studies employed in the study can be found in Supple-mentary Table 1. Data included in all publicly available cohorts were collected according to the respective institutional review board approvals following the human subject protection and data access policies. Written informed consent was obtained from each patient included by the original institutions, and this study was performed following the Declaration of Helsinki. All samples were deidentified and coded following the Health Insurance Portability and Accountability Act (HIPAA) guidelines. No additional ethical approval was needed for this study because all data were obtained from publicly available data.

**Selection of pan-cancer response genes**. Differential gene expression between responder and non-responder patients was computed in nine cohorts of non-BC chemotherapy plus ICI-treated patients (Supplementary Table 1) using the Student's t-test. Then, a score was created by multiplying the $-\log_{10}(p\text{-value})$ by the $\log_2(n)$, where n represents the number of patients per cohort. The score value was converted to negative for downregulated genes. Then, a final score was created by summing the score of each gene

in all datasets, and the 500 genes with the highest absolute score were selected and used as the initial input for the creation of the machine learning classifiers (Supplementary Data 1).

**Construction of a random forest-based classifier.** First, the batch effect between the durvalumab and pembrolizumab I-SPY2 cohorts was checked and corrected using the R/sva package v3.38.0[27]. Gene expression data were normalized to Z-score using R v.4.0.2. The cohort was split into training (60%) and validation (40%) sets. The R/*VarSelRF* v0.7-8[28] package was used to identify the best gene signatures to stratify responder and non-responder patients in the training cohort. We employed *VarSelRF* to remove the least important features in each iteration, selecting the combination of features with the highest predictive potential, as we have previously shown[29–31]. This process was iterated 1,000 times to improve the classifiers, and, in each iteration, the combination of genes with the highest Area Under the Curve (AUC) was selected. Then, the number of iterations in which each gene was included in the best signature was computed to determine the importance of each gene. Finally, the AUC of each combination of the most-repeated genes was calculated. The AUC of the classifier was computed initially in the training cohorts and then validated in independent cohorts using the R/pROC v1.16.2[32] package. The classifier generates a confidence score that can be used to assess the probability of response to ICI in patients with TNBC using a quantifiable measure instead of a dichotomic value of response/no response. Additional molecular signatures known to influence ICI response such as PD-L1 or PD-1 expression and mitotic rate, T-cell, or B-cell signatures were included in the analysis to identify potential improvements in the gene expression-based classifier. These features were merged with the gene expression data and used as input for the RF algorithms to construct hybrid classifiers. The AUC of the hybrid classifiers was computed in the validation cohort. The plots were represented using the R/ggplot2 v3.3.6 and UpSetR v1.4.0 packages.

**Statistics and reproducibility.** All genes with a student's *t*-test *p*-value below 0.05 and an absolute Z-Ratio over 1.5 between responders and non-responders were considered differentially expressed. Overall survival (OS) and disease-free survival (DFS) intervals, the log-rank test p-value, and the risk associated with increments in the gene expression classifier-based scores expressed as the Hazard Ratio were evaluated using the R/*survival* v.3.2-13 package. OS and DFS intervals could not be computed in the I-SPY2 cohort due to the short follow-up of the patients. The Odds Ratio (OR) for the response probability was calculated using the R/*questionr* v0.7.2 package for all available clinical and molecular data, including each signature. The significance of the risk was computed using the Fisher Exact Test (FET). All results were expressed as the OR including the 95% CI.

**Reporting summary.** Further information on research design is available in the Nature Portfolio Reporting Summary linked to this article.

## Results
### Pretreatment differential gene expression profiles in primary TNBC from responder patients.
From the 558 differentially expressed genes, 211 were found upregulated and 347 down-regulated in tumors from ICI plus chemotherapy responder patients (Fig. 1a). These genes showed a good stratification efficiency of patients with TNBC according to the ICI pCR rates (Fig. 1b). This observation indicates the potential of gene expression patterns in predicting ICI response. However, the implementation of the differentially expressed gene signature in a

validation cohort showed a poor performance in discriminating patients based on the response to ICI plus chemotherapy (Fig. 1c). This indicates an overfitting to the discovery cohort and a need for a more complex construction of efficient gene expression-based predictive models.

### A gene expression-based classifier efficiently predicts pCR to ICI in primary TNBC.
To increase the stratification ability of gene expression profiles, we employed random forest, a machine learning algorithm, to select the most informative gene combinations to predict response to ICI in primary TNBC tumors. As expected, the predictive performance in the training cohort for multiple gene combinations was high, even when adding up to 100 genes (AUC$_{val}$ > 0.85; Fig. 2a). Interestingly, we observed an overall good predictive performance in a validation cohort of different classifiers (Fig. 2a). The optimal performance of the signatures was established at 37 genes (Supplementary Table 2). Importantly, a large proportion of these genes is involved in molecular pathways with a potential role in immune response, such as CD8A and CXCL9, cytoskeleton, cell adhesion, metabolism, and transcription regulation (Supplementary Fig. 1). The TNBC ICI response predictive classifier (TNBC-ICI) exhibited a very good efficiency in the validation cohort (AUC = 0.86, CI: 0.65–1, *p* = 0.005; Fig. 2b) and, consequently, in the whole cohort (AUC = 0.89, CI: 0.80–0.98, *p* < 0.001; Fig. 2c, d). Interestingly, in the whole cohort, only one out of nine patients with a score equal to or below 0.4 (*reliable non-responders*) reached a pCR, while 19 out of 20 patients with a score equal to or over 0.65 (*reliable responders*) achieved a pCR, highlighting the relevance of this quantifiable score. However, patients with levels between 0.4 and 0.65 (*uncertain*) showed mixed responses with 38% of the cases reaching pCR. None of the individual genes included in this signature had higher predictive accuracy than the TNBC-ICI classifier (AUC range: 0.5–0.76; Supplementary Fig. 2). This indicates the importance of assessing the expression levels of these genes in the context of the signature constructed by machine learning, in which even genes with individually low to null predictive potential contribute to an efficient predictive signature. In the whole I-SPY2 cohort of ICI-treated patients with TNBC, TNBC-ICI shows a non-significantly superior predictive performance (AUC = 0.89) than the 27-gene signature described by Iwase (AUC = 0.76, *p* < 0.1)[33].

To evaluate whether the TNBC-ICI classifier could be widely employed to predict response to ICI plus chemotherapy in other patients with non-TNBC breast cancer, we evaluated two cohorts of HR + /HER2− tumors from the I-SPY2 trial, one treated with Durvalumab plus olaparib (*n* = 50) and the other treated with Pembrolizumab (*n* = 40). In both cases, we observed a significant but modest predictive performance of the TNBC-ICI classifier (AUC = 0.75, *p* < 0.001 and AUC = 0.72, *p* = 0.01, respectively; Fig. 2e, f).

### The accuracy of TNBC-ICI does not improve by integrating other molecular signatures.
The predictive potential of the signature was compared with other molecular signatures associated with ICI response. These included the T-cells, B-cells, mast cells, dendritic cells, mitotic rate, PD-L1, and PD1, among other signatures. In all the cases, the TNBC-ICI classifier displayed a higher predictive potential than all specific molecular signatures (Supplementary Fig. 3). We, therefore, employed a similar machine learning-based strategy to combine the ICI-related molecular signatures and generate an accurate predictive classifier. However, the resulting classifiers showed poor predictive performance for the validation cohort (AUC = 0.67 CI: 0.4–0.95, *p* = 0.1; Fig. 3a, b). Thus, to identify potential synergism between

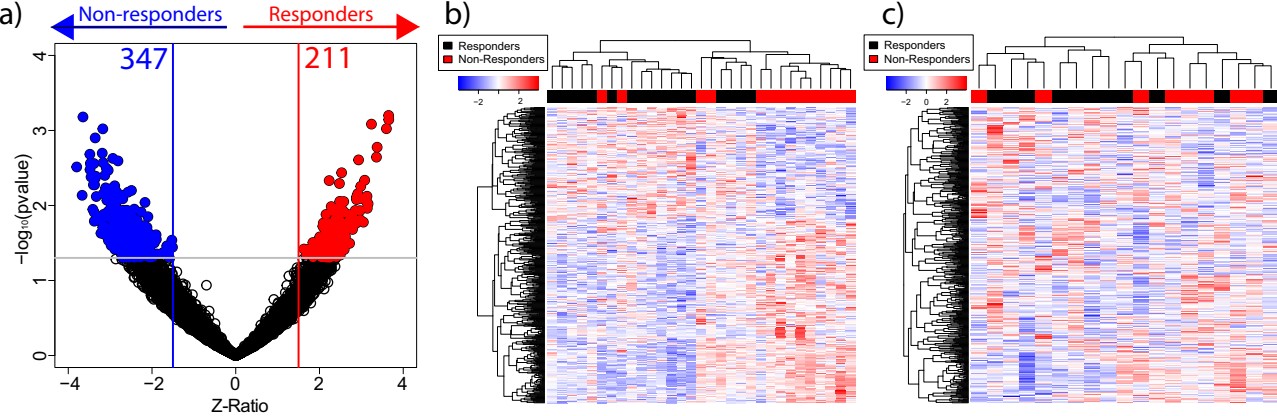

**Fig. 1 Analysis of differentially expressed genes in primary TNBC in response to ICI plus chemotherapy. a** Volcano plot representing the differential gene expression in Z-ratio (X-axis) and the t-test p-value, expressed as –log₁₀(p-value) (Y-axis). **b** Heatmap representing the hierarchical clustering using all differentially expressed genes in the whole cohort and (**c**) in the validation cohort. Differentially expressed genes can successfully cluster responder and non-responder patients when using the whole cohort, but not in the validation cohort.

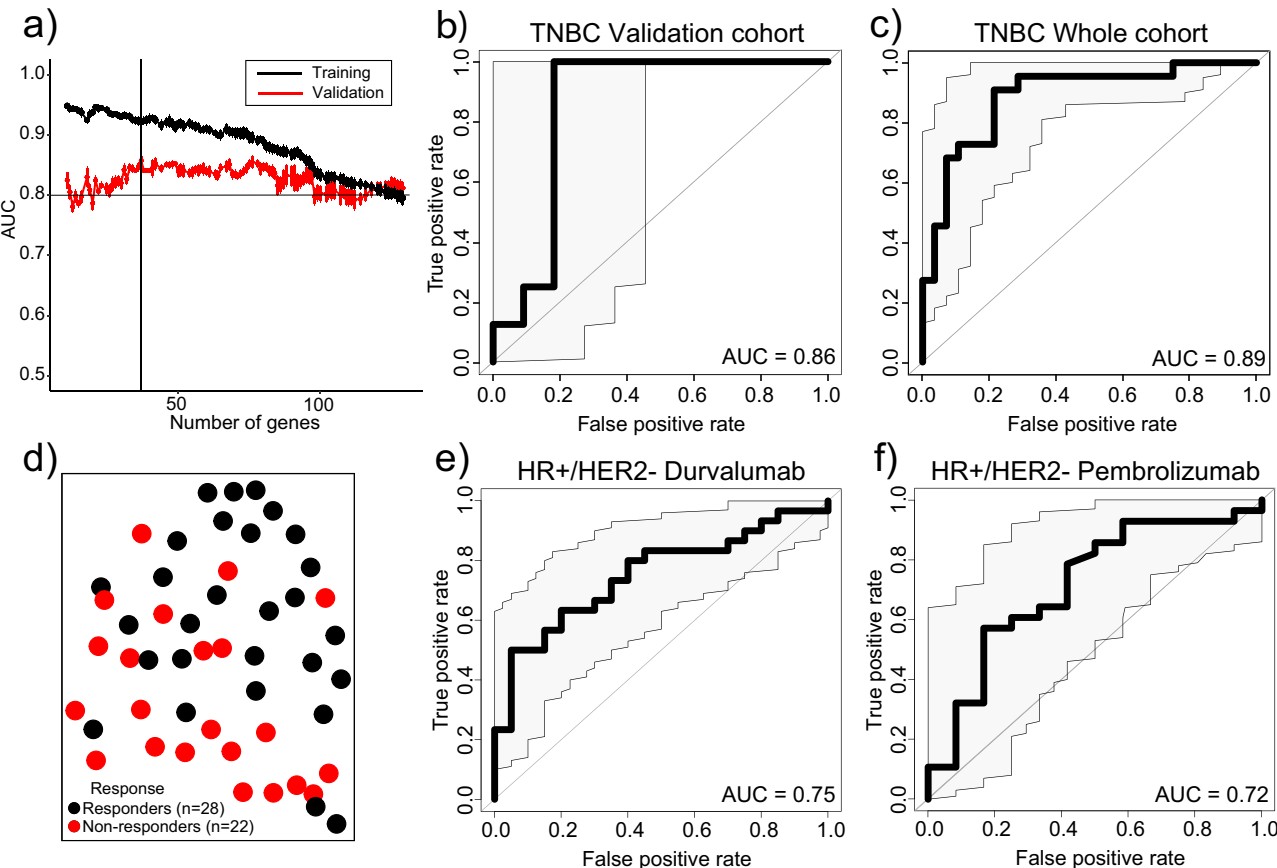

**Fig. 2 Generation of a gene signature to predict the pathological complete response (pCR) to immune checkpoint inhibitors (ICI) plus chemotherapy in primary TNBC. a** Change in the Area Under the Curve (AUC) after decreasing the number of used genes to predict response. The lines represent the mean AUC ± SD ($n = 1000$). **b, c** ROC curve representing the AUC of TNBC-ICI in patients with TNBC in the (**b**) validation ($n = 19$) and the (**c**) whole cohort ($n = 50$). **d** Uniform Manifold Approximation and Projection (UMAP) displaying clustering of patients with TNBC using the genes in TNBC-ICI. **e, f** ROC Curve representing the AUC of TNBC-ICI in breast cancer HR + /HER2− patients treated with (**e**) Olaparib plus durvalumab ($n = 50$) and (**f**) pembrolizumab ($n = 40$). The shaded area represents the confidence interval with a 95% confidence level.

the TNBC-ICI classifier and ICI-related molecular signatures, we generated hybrid nomograms using machine learning to combine the signatures with our classifier. The hybrid nomogram showed a modest performance in identifying responder patients (AUC = 0.75, CI: 0.51–0.99, $p = 0.04$), still below the

performance shown by the TNBC-ICI classifier alone (AUC = 0.86 CI: 0.65–1, $p = 0.005$; Fig. 3a). These findings were maintained in the whole cohort (AUC_signatures = 0.72, AUC_hybrid = 0.83, AUC_TNBC-ICI = 0.89; Fig. 3B). We measured the association of each molecular signature and the TNBC-ICI classifier with the

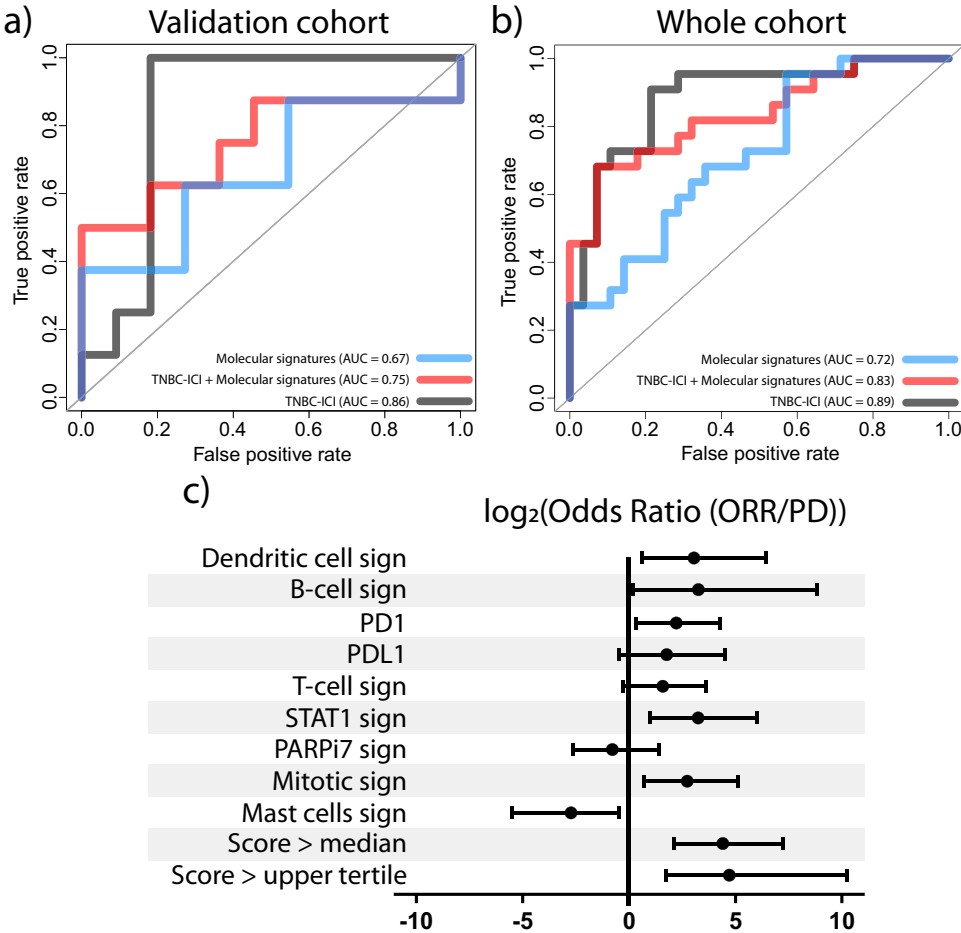

**Fig. 3 Comparison of efficacy of TNBC-ICI and other molecular signatures.** ROC Curves displaying the performance of TNBC-ICI (black) compared with the performance of other molecular parameters-based signatures (blue) and hybrid nomograms (red) in the (**a**) validation cohort ($n = 19$) and the (**b**) whole cohort ($n = 50$). **c** Forest plot displaying the Odds Ratio (OR) (±95% confidence interval) of pathological complete response (pCR) to immune checkpoint inhibitors (ICI) for different molecular signatures, including the score of TNBC-ICI in primary TNBC ($n = 50$). The TNBC-ICI classifier score signature has a higher OR for pCR than any molecular signature.

likelihood of responding to ICI plus chemotherapy. Patients with a high TNBC-ICI classifier score have a significantly higher probability of achieving a pCR (OR 21.3, 95% CI: 4.3–151.5, $p < 0.001$), outperforming all other molecular parameters (Fig. 3c).

**The TNBC-ICI classifier is specific for patients with TNBC treated with ICI.** We evaluated associations between the score of the TNBC-ICI classifier and survival intervals for patients with TNBC that have been treated with chemotherapy but not with ICI. We found that the TNBC-ICI classifier score is not associated with either DFS (Fig. 4a) or OS (Fig. 4b) in ICI-naïve patients. Furthermore, it does not predict response to chemotherapy in the I-SPY2 cohort (AUC = 0.53). This finding suggests that TNBC-ICI identifies ICI-sensitive tumors and not a subset of patients with TNBC with intrinsic less aggressive disease. We tested the TNBC-ICI classifier in patients with other solid tumors treated with ICI plus chemotherapy. The TNBC-ICI classifier showed a poor overall predictive performance in non-TNBC tumors (median AUC = 0.67; Supplementary Fig. 4). Yet, we observed: (i) a significant performance in predicting ICI response in patients with locally advanced, unresectable melanomas (AUC = 0.68; $n = 48$; $p = 0.01$), metastatic melanoma (AUC = 0.73; $n = 26$; $p = 0.02$), or early esophageal cancer (AUC = 0.73, $n = 35$, $p = 0.01$); (ii) a significant, but potentially clinically irrelevant ICI

response predictive performance for patients with bladder cancer (AUC = 0.62; $n = 348$; $p < 0.001$) and renal cancer (AUC = 0.65; $n = 56$; $p = 0.03$); and (iii) a non-significant performance for patients with locally advanced esophageal cancer (AUC = 0.53; $n = 37$; $p = 0.4$). No significant differences were observed between the classifier accuracy of cancer types considered immune hot (melanoma, renal) or immune cold (bladder, esophageal). This observation reflects the fact that most of the genes in the TNBC-ICI classifier are involved in immunity, not only in the TNBC context but different tissue types.

**Discussion**

Immunotherapy is altering the way TNBC is treated and demonstrating long-lasting responses; nevertheless, many individuals do not respond and, despite being mild, there is still a possibility for negative side effects. Importantly, response rates to ICI in unselected patients with TNBC are still below those achieved in other cancers such as melanoma or lung carcinoma. Biomarkers relevant for the selection of patients with other cancers who are likely to respond to ICI have shown poor or modest performances in predicting response to ICI in patients with TNBC[34,35]. Thus, there is a significant need to identify TNBC-specific markers that predict response to ICI. Here, we employed gene expression data of 721 patients treated with ICI plus chemotherapy to construct and evaluate TNBC-ICI, a gene

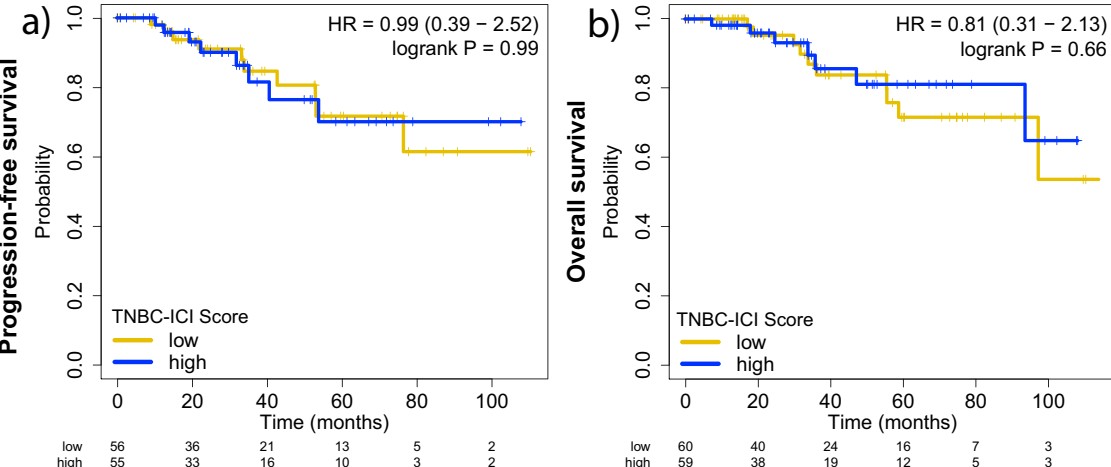

**Fig. 4 Survival analysis in immune checkpoint inhibitors (ICI)-naïve patients with TNBC.** Kaplan-Meier curves displaying the (**a**) Disease-Free Survival (DFS) and (**b**) Overall Survival (OS) of ICI-naïve patients with TNBC, stratified using TNBC-ICI. The classifier does not predict increased survival in ICI-naïve patients with TNBC.

expression-based classifier that predicts pCR to ICI plus chemotherapy in TNBC. This classifier includes 37 genes and shows better predictive performance than other relevant molecular parameters. For instance, patients with TNBC with a score in the upper tertile were 21.3 more likely to achieve a pCR to ICI plus chemotherapy than those with lower TNBC-ICI scores (lower and middle tertile), while other studies show inferior performances using other biomarkers such as PD-L1[7,15]. However, an important caveat to consider when interpreting these results is the fact that the evaluation of the TNBC-ICI is limited to patients in the current set that we have randomly split into independent training and validation cohorts. Overall, these findings suggest better performance of gene signature classifiers, especially those selected by artificial intelligence algorithms than single biomarkers. Our study shows that TNBC-ICI has a high accuracy in predicting response in TNBC tumors with both high ($\geq$0.65, 95% accuracy) and low ($\leq$0.4, 89% accuracy) scores, but this predictive performance is lower in cases with scores between these cutoffs (0.4 to 0.65; eight responders and 13 non-responders). Thus, we propose that the outcomes can be categorized into three groups: reliable responders, uncertain cases, and reliable non-responders, with potential clinical relevance. It is worth noting that further investigation is needed to refine and improve the TNBC-ICI accuracy, particularly for cases with average scores. Our results emphasize the importance of using TNBC-ICI in conjunction with clinical judgment to make informed treatment decisions. Yet, the relatively small number of cases in our study and the consequent wide confidence intervals are limitations that require validation in larger prospective studies to confirm the predictive accuracy of the TNBC-ICI classifier. Thus, it is essential to validate these findings in independent cohorts to ensure the reliability and generalizability of the classifier.

Our study is not the first to generate ICI predictive models based on gene expression. Yet, our approach, involving machine learning to identify gene signatures for ICI response, is different from other studies. Other complementary ICI predictive markers have been established based on predetermined immune-related tumor functions or using non-supervised algorithms to create subgroups with differential responses to ICI[35]. While some of these studies included genes in the classifier according to previous knowledge about the molecular function of each gene, we employed machine learning methods to systemically select the features with the highest predictive value regardless of the gene function. However, a significant proportion of the genes in the classifier were involved in

immunity, highlighting the relevance of the immune microenvironment in the response to ICI. Other genes were related to cell adhesion, metabolism, cell cycle, and transcription regulation (Supplementary Table 2). Additionally, we have used data from nine non-BC datasets (Supplementary Table 1) to identify pan-cancer differentially expressed genes in responder and non-responder patients to increase the accuracy of the classifier. Furthermore, the random forest algorithm assesses the feature importance of each gene alone and in the context of a signature, reducing information redundancy, and thus increasing the classifiers' accuracy above previous studies (OR: 21.3 vs 1.6–1.7)[35].

Therefore, for patients with TNBC, the TNBC-ICI signature displays a higher accuracy than pan-cancer subtyping methodologies and classifiers created using genes with potentially relevant functions. In their study, Bagaev et al. showed four pan-cancer conserved microenvironment subtypes (Immune-Enriched Fibrotic, Immune-Enriched Non-Fibrotic, Fibrotic, and Desert) with a significant predictive value for immunotherapy response on different tumor types (melanoma, bladder, and lung cancer)[34]. Yet, due to the lack of breast cancer specimens and variations in the feature selection methodology, a direct comparison with the TNBC-ICI cannot be performed. In another study of the same group, machine learning was applied to reconstruct the tumor microenvironment using bulk RNA-seq data. This model, combined with the TMB and PD-L1 levels predicts response to immunotherapy in non-breast cancer tumors (bladder, gastric, kidney, and melanoma; AUC = 0.75). This is similar to the observed accuracy when applying TNBC-ICI to patients with HR+ breast cancer treated with pembrolizumab and durvalumab (AUC = 0.72, $p = 0.01$ and AUC = 0.75, $p < 0.001$, respectively) and to patients with non-breast cancer (median AUC = 0.67). Other subtype-based signatures have revealed similar results in TNBC (OR: 2.9–5.9) and HR+ (OR = 5.4) patients treated with ICI[36], while signatures based solely on chemokines displayed significant, but more modest results in ICI-treated patients with TNBC (OR: 2.5–4)[37]. The ImPrint immune signature, developed by Mittempergher et al., displays a comparable accuracy to TNBC-ICI in ICI-treated BC patients, including TNBC, showing 89% sensitivity and 58% specificity in these patients[38]. A recent study shows the significant power of a model combining gene expression of immune-related genes, including CD274 and PDCD1, and the TMB to identify pan-cancer ICI-treated patients that will have a longer PFS and OS[39].

Employing machine learning, other studies have identified similar performances in different types of cancers such as melanoma or gastrointestinal cancer[40–42]. We also compared the

performance of TNBC-ICI with the 27-gene signature from Iwase et al. and we observed non-significant differences in the predictive efficiency (AUC: 0.89 vs 0.76, respectively)[33]. Although different data types have shown an increase in the stratification capacity in other cancers[43], we found that integrating other molecular signatures has not shown an improvement in the performance of the gene expression-based classifier. This event could be related to a "*plateau effect*" of the prediction performance due to the high efficacy of TNBC-ICI. Moreover, additional benefits could be obtained from studying other facets of tumor response, such as tumor shrinking. However, these parameters could not be analyzed due to the lack of available data in the studied datasets.

Showing that TNBC-ICI is specific to BC disease, it displayed a high accuracy in TNBC (AUC = 0.86) and a modest, still significant prediction of ICI response in HR+/HER2− BC patients (AUC: 0.72–0.75), but poor accuracy in non-BC tumors (median AUC = 0.67). These findings suggest that, while the classifier was constructed and trained to predict ICI response in TNBC tumors, the included gene list may be involved in ICI response in different tissue contexts. The pursuit of a pan-ICI response predictive system would require larger datasets to control for differences in treatment, tissue types, and other clinical and demographic variables. Unfortunately, due to the lack of accessibility to the datasets, we could not test the classifier in tumors from lung or head and neck carcinomas or additional TNBC cohorts. Furthermore, the accuracy in HR+/HER2− BC patients is substantially lower than in patients with TNBC, limiting the effective use of TNBC-ICI to patients with TNBC. However, the significant but limited accuracy of the classifier in this group of patients serves as a demonstration of the potential of TNBC-ICI to predict response to ICI. Moreover, the TNBC-ICI classifier was not significantly associated with survival intervals or higher rates of response in patients with TNBC treated with chemotherapy alone, suggesting the identification of tumor features involved in response to ICI and not a sub-group of patients with intrinsic extended survival.

While single biomarkers adapted to routine pathology assays are still the gold standard for fast and cost-effective specimen diagnosis, there is still significant room for improvement. One advantage of the molecular signatures classifiers is the accuracy and reproducibility of the methods. While other methods such as immunohistochemistry can be biased by different factors, including the antibody type, lots, reagents, and pathologists reading the results, transcriptome analysis, while still having a certain batch effect, reduces the "human effect" on the analysis of the results[44]. However, each approach has applications in the appropriate diagnostic setting.

In summary, integrating gene expression profiles of TNBC specimens using artificial intelligence provides a robust gene expression-based classifier that can predict pCR to ICI plus chemotherapy treatment in TNBC. This classifier, called TNBC-ICI, exhibits promising predictive potential based on our study but must be tested in independent TNBC cohorts to ensure the reproducibility of the results in patients with different demographical and clinical features to allow for a precise selection of patients who will respond to ICI treatment. A limitation of the application of the TNBC-ICI classifier is the scarcity of datasets containing reliable transcriptome and outcome data from patients with TNBC treated with ICI plus chemotherapy, which limits its extended validity. Nonetheless, we have opened a simplified version of the code at the GitHub repository that can be easily employed and improved by other researchers. We plan to update the TNBC-ICI model as new transcriptomic and clinical data become available, to enhance its accuracy and its utility as a predictive tool for treatment response. Updated versions of the classifier will be made available in the GitHub repository (see Code Availability section).

## Data availability

All the data employed in the study has been obtained from publicly available databases[24,25,45–52], including the Gene Expression Omnibus (GEO), ArrayExpress, and directly from the IMvigor210CoreBiologies R package. All the accession numbers are available in Supplementary Table 1. Source data for the figures are available at the Zenodo repository[53].

## Code availability

All the code employed in the study is available at GitHub (https://github.com/mensenyat/TNBC-ICI) and Zenodo[54] repositories. These files include a simplified version of the code that can be employed to check the validity of the method and a dataset and code that can be used to apply TNBC-ICI to new samples. This code will be updated in GitHub with data from additional public datasets when they are released.

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

## Acknowledgements

This work was supported by Instituto de la Salud Carlos III (ISCIII) Miguel Servet II (CPII22/00004) and Sara Borrell (CD22/00026) contracts, and AES2022 (#PI22/01496), co-funded by the European Union, the Asociación Española Contra el Cancer (AECC), the "Liberi" program (Health Research Institute of the Balearic Islands, IdISBa), the department of European Funds, University and Culture of the Government of the Balearic Islands (FPI/037/2021), the "CONTIGO Contra el Cancer de Mujer" foundation (#MERIT project), the Fashion Footwear Association of New York (FFANY) Foundation, the UCLA Breast Epigenetics Program, and the Epi-UCLA22 project.

## Author contributions

D.M.M., M.E.M., and J.I.J. conceived of the study. M.E.M., D.M.M., J.I.J., and P.L.A. wrote the first draft of the manuscript. M.P.S. contributed to the design of computational biology analyses. M.E.M., P.L.A., and S.I.M. performed data analysis and visualization. M.L.D., J.L.B., J.C., and M.M., suggested clinical and immunological contextualization of the results. M.E.M., J.I.J., J.L.B., M.M., M.L.D, J.C, and D.M.M. contributed to the interpretation of the data and critical revision of the manuscript drafts. All authors have reviewed and approved the final version of the manuscript.

## Competing interests

The authors declare no competing interests.
