## [Peer Review File · Communications Medicine]

Reviewers' comments:

Reviewer #1 (Remarks to the Author):

Authors deduced a gene expression signature to predict immunotherapy response in TNBC patients. Below are my major concerns.

- The work does not benchmark against the array of recent works produced by Boston Gene (<https://bostongene.com/about-us/publications/>; see their cancer cell papers like: [https://www.cell.com/cancer-cell/fulltext/S1535-6108\(21\)00222-1](https://www.cell.com/cancer-cell/fulltext/S1535-6108(21)00222-1)).

- Survival analysis results are largely inconclusive.

- None/Few biological insights were presented, reducing the work to bioinformatics analysis.

As such, I couldn't be more positive in this instance.

Reviewer #2 (Remarks to the Author):

The authors consider the construction of a machine-learning-based binary classifier to predict response to immunotherapy, via immune checkpoint inhibitors (ICI), for primary triple negative breast cancer (TNBC). TNBC is well known for its poor prognosis and the relative absence of treatment methods. Predicting response to ICI is an important problem which is being considered by a number of groups. Apart from enhancing treatment, identification of patients who respond is of great interest to pharma because it changes the economics of drug administration. Response to ICI has been variable on a patient and cancer-type basis. For other types of cancers, such as small cell lung cancer, spread in response to ICI is quite substantial ranging from no effect at all to complete remission. Thus the objective of the study is well posed.

There have been various studies of the impact of ICI in the treatment of advanced TNBC and the results have been variable. Some studies have suggested a median overall increase in survival of up to six months in those receiving ICI, over chemotherapy alone. However, TNBC is likely to be a scenario where a reasonable response to immunotherapy could be expected because of tumour infiltration by lymphocytes and typically higher PD-L1 expression. It is also the case that TNBC patients have a typically

higher number of driver variants in the genome, compared with other breast cancer types, which *may* indicate more neo-antigens and possibly greater effects from ICI.

In terms of competing literature, there has been significant fairly recent interest in understanding/predicting response to ICI in TNBC (mostly published 2021/2022). Much of this has been oriented towards finding biomarkers, some has been gene signatures, and there are a few others, e.g. using unsupervised learning. I've not seen supervised learning used to date in this context.

The background of the reviewer is machine learning (ML) and bioinformatics, and hence the focus of the comments are towards the machine learning.

Focussing on the machine learning, there are a number of issues which would need addressing:

1. One overall comment is that there is a sometimes slightly confusing usage of the terms 'training', 'validation' and 'testing', at least from the viewpoint of a ML reader. The training data is used to train the classifier, validation data is used to find hyperparameters within the classifier, or maybe a stopping point for feature selection. After training is complete you then evaluate on unseen test data.
2. Page 5, lines 9,10 and 11: some aspects of these sentences are not clear. In particular, deep learning is commonly regarded as a part of ML, and it is un-clear what is meant by the phrase 'integration of several layers of information', which suggests thinking of a deep learning architecture, but the subsequent sentence suggests more a consideration of data integration methodologies from ML. Possibly a better phrasing of the sentences might be: 'The development and adaptation of machine learning methods, such as deep learning, allows for extraction of informative features within the layers of a neural network, and deep learners, and other machine learning methods, allow for the integration of disparate data sources from imaging, clinical covariates, histology and molecular profiling.' (?)
3. Comment (p.6): Data used for the study is described on page 6 and is publicly available data from the Gene Expression Omnibus. Gene expression profiling was selected from this source and the binary classification considered was based on pCR (and NO), defining the absent class of invasive cancer, and those not fitting into this class are classified as non-responders. This is a very basic binary classification: I know researchers who have been looking at predicting response to ICI using machine learning have been trying to establish more refined criteria for measuring response, inclusive of observed measures of tumor shrinkage. The authors also use data from The Cancer Genome Atlas.
4. Page 6, lines 14-15. A sentence following in this paragraph would need clarification in the Supplementary to ensure reproducibility of all statements. Specifically this states 'Clinical annotations were manually curated using the scan clinical reports to solve missing or discrepant data issues'. To ensure reproducibility the criteria used here should be described as closely as possible in the Supplementary materials.
5. Page 7, lines, 8-9. The authors used an iterative feature elimination algorithm to select those features which are informative for the stated classification task, iteratively removing the least important features. This aspect of the study is not described in sufficient detail. In particular as previously noted, data is typically split into training, validation and test data. In the current description there is no clear criterion given for the stopping point for the removal of features and this can substantially affect the model. Commonly the classifier, for the given feature set, is evaluated on validation data, and feature selection is stopped when the validation accuracy is optimal: page 7, line 13 does suggest this is the

approached used though. The methods used have to be properly described in the Supplementary so that the criterion for removing features is described along with the stopping point for the selection of features. The package may also not be optimal in that methods which remove the least important features, one at the time, can be a little bit less accurate than other algorithms which can remove features, but have a move to possibly reintroduce them at some later stage (rather like tabu list methods from optimization theory). Two comments: (a) the features/gene found via feature selection may not be very biologically relevant, they can be useful for prediction but give no biological insight (thus you can find the genes are associated with altered metabolic rates and not pathogenicity); (b) I'm not suggesting the authors do this, but you can randomly partition the data into two separate independent datasets, then pursue the feature selection process independently on both, the coincidence that you find the same features/genes in common in both is then governed by the hypergeometric distribution (with an associated adjusted p-value).

6. Page 9, line 17. The authors report that the error drops when using less than 30 genes. It is worth checking with the training error may now be increasing because, when the number of gene/feature set is too small, you are in a low dimensional space. Comment: another feature which is lacking in this coverage of machine learning is whether a confidence measure can be applied to the predicted label: this makes the use of machine learning much more powerful in that a clinician might be inclined to treat high confidence prediction cases but would not do so if the confidence is low.

7. The discussion in later parts of the paper appears reliable. The comments that they observe a significant but modest predictive performance of the TNBC – ICI classifier is generally in line with expectations from previous studies predicting response to ICI, in terms of possible AUC, for other cancer types where response has been noted.

8. Page 10, line 7. Comment: you have to be careful with the integration of molecular signatures. These may have been derived from data mentioned in the study and could lead to circularity (i.e. an unfair data leakage which could bias the predictor towards over-stated test accuracies because relevant information about the test label is presented to the classifier). This is very unlikely to have happened here since later (I17+, etc), the authors report modest effects when integrating molecular signatures.

9. Comment: (not suggesting the authors do this but it is an observation about dataset sizes and possible 'plateau effects', line 10, page 13): all machine learning algorithms follow a Zipf's Law dependence of the empirical test error, z , on the sample size, m , via $z = a m^{-\alpha} + b$ where α is related to the efficiency of the algorithm. If possible, you can train at different m , find z and then use this, and optimization, to find a , b and α . A non-zero b as $m \rightarrow \infty$ can indicate an intrinsic prediction limitation, possibly noise or missing information, but can give insight as to whether much more useable prediction accuracies can be achieved, if more data could be acquired.

10. *Making user-friendly code and a web-interface available would greatly increase support for publishing this paper*.

The above comments and suggestions are for the purposes of clarity and possible improvements (I'm not suggesting the authors necessarily go with any comments).

Overall, TNBC is a problematic disease with limited current treatments and generally poor outcomes. Hence, given the relative absence of supervised learning methods for ICI response for TNBC, this would be a good supporting case for this paper.

Reviewer #3 (Remarks to the Author):

Summary: The manuscript provides an example of one approach for developing a gene classifier for prediction of clinical outcomes following chemotherapy plus ICI in TNBC.

Major concerns:

- As discussed in page 9 starting at line 12, the validation cohort was used to tune the classifier. Therefore, additional validation in an external cohort is necessary. Perhaps it is possible to collaborate with investigators of the neoTRIP or NeoPACT or another trial.
- Limited information is provided about how the classifier was compared to other leading gene classifiers, some which are now clinically available (i.e. 27-gene IO score is mentioned briefly in the discussion page 13 line 5.) Please provide a more comprehensive summary of how the comparison was made and include in the results section rather than mentioning it only in the discussion. Also please provide additional summary about other gene classifiers in the literature, including the Agendia Impact classifier developed by ISPY2 investigators (SABCS 2022), and others (for example, the chemokine 12 score presented by Soliman et al at SABCS 2021)
- Most classifiers are being developed as a prediction biomarker, i.e. to predict benefit of adding ICI to chemo using pCR as the clinical outcome, comparing the chemo v. ICI+chemo arms. The gene classifier in this manuscript is trained as a prognostic biomarker, i.e. to forecast pCR outcome of ICI+chemo. Please justify this approach, and provide more clarity on how then the biomarker would be clinically useful, especially in comparison to the 27-gene IO score which is shown to predict ICI benefit when added to chemo.

Minor concerns:

- Additional grammatical revisions/refinement are necessary before consideration of publication.
- Page 4 line 16 – atezo did not meet OS endpoint in overall cohort but there was an OS benefit in PD-L1+ cohort (although this was not a prespecified endpoint per the statistical analysis plan).
- Please provide a plausible next step for your investigative plan with the classifier

Reviewers' comments

Reviewer #1 (Remarks to the Author):

Authors deduced a gene expression signature to predict immunotherapy response in TNBC patients. Below are my major concerns.

The work does not benchmark against the array of recent works produced by Boston Gene (<https://bostongene.com/about-us/publications/>; see their cancer cell papers like: [https://www.cell.com/cancer-cell/fulltext/S1535-6108\(21\)00222-1](https://www.cell.com/cancer-cell/fulltext/S1535-6108(21)00222-1)).

Response to Reviewer: Thank you for pointing this out. A comparison with the methodology employed and models generated by Boston Gene in their two Cancer Cell papers, and the accuracy of their system to predict immunotherapy response has been included in the revised version of our manuscript (**Page 13, Line 5**). Since their studies did not include breast cancer patients and the methodology for generating the models is different, a fair comparison with the TNBC-ICI cannot be conclusive.

Survival analysis results are largely inconclusive.

Response to Reviewer: We agree with this observation. Due to the design of the trials included in our study and the time from initial follow-up on these early stages of the TNBC disease, a large proportion of the immunotherapy-treated patients lack of disease-related events to report. Thus, the survival analyses in our study are limited to chemotherapy-treated TNBC patients with extended follow-up. However, this analysis shows that the TNBC-ICI signature does not identify tumors with a low risk of relapse, and therefore a potentially better response to treatment, but tumors with a higher likelihood of responding to immunotherapy. We have highlighted this observation in our revised manuscript (**Page 14, Line 18**)

None/Few biological insights were presented, reducing the work to bioinformatics analysis.

Response to Reviewer: Thank you for pointing this out. This is an interesting observation that shows the agnostic nature of the construction of the predictive classifier. The aim of the study is to identify an efficient gene signature that predicts the response to ICI regardless of the molecular function of each gene. In addition, when applying machine learning techniques, it is commonly observed that redundant information such

as more than one gene per pathway or molecular function (highly correlated features) is excluded. However, in the final TNBC-ICI signatures, we have identified an over-representation of immune-related genes (**Suppl. Fig. 1**). We have commented on this in our revised manuscript (**Page 12, Line 27**).

As such, I couldn't be more positive in this instance.

Response to Reviewer: Thank you again for revising the manuscript and for the constructive criticism.

Reviewer #2 (Remarks to the Author):

The authors consider the construction of a machine-learning-based binary classifier to predict response to immunotherapy, via immune checkpoint inhibitors (ICI), for primary triple negative breast cancer (TNBC). TNBC is well known for its poor prognosis and the relative absence of treatment methods. Predicting response to ICI is an important problem which is being considered by a number of groups. Apart from enhancing treatment, identification of patients who respond is of great interest to pharma because it changes the economics of drug administration. Response to ICI has been variable on a patient and cancer-type basis. For other types of cancers, such as small cell lung cancer, spread in response to ICI is quite substantial ranging from no effect at all to complete remission. Thus the objective of the study is well posed.

*There have been various studies of the impact of ICI in the treatment of advanced TNBC and the results have been variable. Some studies have suggested a median overall increase in survival of up to six months in those receiving ICI, over chemotherapy alone. However, TNBC is likely to be a scenario where a reasonable response to immunotherapy could be expected because of tumour infiltration by lymphocytes and typically higher PD-L1 expression. It is also the case that TNBC patients have a typically higher number of driver variants in the genome, compared with other breast cancer types, which **may** indicate more neo-antigens and possibly greater effects from ICI.*

In terms of competing literature, there has been significant fairly recent interest in understanding/predicting response to ICI in TNBC (mostly published 2021/2022). Much of this has been oriented towards finding biomarkers, some has been gene signatures, and there are a few others, e.g. using unsupervised learning. I've not seen supervised learning used to date in this context.

The background of the reviewer is machine learning (ML) and bioinformatics, and hence the focus of the comments are towards the machine learning.

Focussing on the machine learning, there are a number of issues which would need addressing:

- 1- One overall comment is that there is a sometimes slightly confusing usage of the terms 'training', 'validation' and 'testing', at least from the viewpoint of a ML reader. The training data is used to train the classify,*

validation data is used to find hyperparameters within the classifier, or maybe a stopping point for feature selection. After training is complete you then evaluate on unseen test data.

Response to Reviewer: Thanks for highlighting this issue. This has been corrected in the revised version of the manuscript.

2- Page 5, lines 9, 10 and 11: some aspects of these sentences are not clear. In particular, deep learning is commonly regarded as a part of ML, and it is un-clear what is meant by the phrase 'integration of several layers of information', which suggests thinking of a deep learning architecture, but the subsequent sentence suggests more a consideration of data integration methodologies from ML. Possibly a better phrasing of the sentences might be: 'The development and adaptation of machine learning methods, such as deep learning, allows for extraction of informative features within the layers of a neural network, and deep learners, and other machine learning methods, allow for the integration of disparate data sources from imaging, clinical covariates, histology and molecular profiling.' (?)

Response to Reviewer: Thank you very much for this suggestion. We have added it to the manuscript and it significantly clarifies the explanation of the approach employed in the study (**Page 5, Line 8**).

3- Comment (p.6): Data used for the study is described on page 6 and is publicly available data from the Gene Expression Omnibus. Gene expression profiling was selected from this source and the binary classification considered was based on pCR (and N0), defining the absent class of invasive cancer, and those not fitting into this class are classified as non-responders. This is a very basic binary classification: I know researchers who have been looking at predicting response to ICI using machine learning have been trying to establish more refined criteria for measuring response, inclusive of observed measures of tumor shrinkage. The authors also use data from The Cancer Genome Atlas.

Response to Reviewer: We agree with this observation and believe in the importance of considering the partial response and even stable disease when predicting response

to immunotherapy. However, in this study, we were limited by the clinical data provided by the clinical studies. As such, the only parameter that was uniformly reported was pCR. As recommended by the guidelines to evaluate response to neoadjuvant therapies, we established pCR as ypT0/is, ypN0 (Earl H, BMC Medicine 2015), which represents the absence of disease in the primary site as well as in the locoregional lymph nodes. In our revised manuscript, we have indicated this limitation and the importance of considering other clinical responses to immunotherapy, such as tumor shrinkage or increased survival (**Page 14, Line 2**).

4- Page 6, lines 14-15. A sentence following in this paragraph would need clarification in the Supplementary to ensure reproducibility of all statements. Specifically this states 'Clinical annotations were manually curated using the scan clinical reports to solve missing or discrepant data issues'. To ensure reproducibility the criteria used here should be described as closely as possible in the Supplementary materials.

Response to Reviewer: Thanks for pointing this out. We have clarified this approach to ensure reproducibility. The guidelines employed are the American Society of Clinical Oncology/College of American Pathologist (ASCO/CAP) guidelines to identify misclassified samples, i.e., the definition of TNBC tumors in cases with the discrepancy on the hormone receptors or HER2 expression statuses/levels (**Page 6, Line 16**).

5- Page 7, lines, 8-9. The authors used an iterative feature elimination algorithm to select those features which are informative for the stated classification task, iteratively removing the least important features. This aspect of the study is not described in sufficient detail. In particular as previously noted, data is typically split into training, validation and test data. In the current description there is no clear criterion given for the stopping point for the removal of features and this can substantially affect the model. Commonly the classifier, for the given feature set, is evaluated on validation data, and feature selection is stopped when the validation accuracy is optimal: page 7, line 13 does suggest this is the approached used though. The methods used have to be properly described in the Supplementary so that the criterion for removing features is described along with the stopping point for the selection of features.

The package may also not be optimal in that methods which remove the least important features, one at the time, can be a little bit less accurate

than other algorithms which can remove features, but have a move to possibly reintroduce them at some later stage (rather like tabu list methods from optimization theory).

Response to Reviewer: Thank you for these suggestions. As you mentioned, we used the validation cohort to select the optimal number of features. The package employed to perform this task (R/*VarSelRF* v0.7-8) does indeed remove one feature in each step without the possibility of re-introduction, thus, as you mention, removing some combinations that could have high accuracy. However, as this same process is repeated 1,000 times, a high-performing combination of features initially excluded can be selected in subsequent iterations. In addition, to minimize the chances of missing informative combinations due to this approach, we rank the genes based on the number of times that each gene has been selected as a final classifier and perform a Random Forest using this list. Furthermore, the initial selection of the genes was computed using the pan-cancer cohorts, selecting the genes with differential expression between responders and non-responders to immunotherapy. Even though we know that this method is not perfect, it displayed better results than employing directly the genes obtained from the *R/VarSelRF* algorithm. This part has been rewritten to improve the readability of the section and the reproducibility of the method (**Page 7, Line 11**).

Two comments: (a) the features/gene found via feature selection may not be very biologically relevant, they can be useful for prediction but give no biological insight (thus you can find the genes are associated with altered metabolic rates and not pathogenicity); (b) I'm not suggesting the authors do this, but you can randomly partition the data into two separate independent datasets, then pursue the feature selection process independently on both, the co-incidence that you find the same features/genes in common in both is then governed by the hypergeometric distribution (with an associated adjusted p-value).

Response to Reviewer: We totally agree with this observation. While the biological functions altered affected on non-responder tumors are of great value for potential therapeutical developments or changes in the clinical management of the patients, this study aimed to construct highly predictive classifiers. However, we have identified an over-representation of immune-related genes on the final TNBC-ICI classifier that may be reflecting, in part, molecular functions that are relevant for the immune response on

TNBC (Suppl. Fig. 1). We have commented on this in our revised manuscript (**Page 12, Line 27**).

6- Page 9, line 17. The authors report that the error drops when using less than 30 genes. It is worth checking with the training error may now be increasing because, when the number of gene/feature set is too small, you are in a low dimensional space.

Response to Reviewer: We are sorry about the confusion, the error rate in the validation cohort “increases” when using less than 30 genes. However, the error rate in the training cohort is indeed decreased when using a really small number of genes in this type of study, possibly due to the potential overfitting of the classifier. We have clarified this in our revised manuscript (**Page 9, Line 17**)

7- Comment: another feature which is lacking in this coverage of machine learning is whether a confidence measure can be applied to the predicted label: this makes the use of machine learning much more powerful in that a clinician might be inclined to treat high confidence prediction cases but would not do so if the confidence is low.

Response to Reviewer: Yes, the classifier indeed generates a *confidence score* of the predicted label. The employed algorithm produces a score from 0 to 1 for the Responder and for the Non-Responder group, which can be employed by the clinician to select high-confidence and low-confidence cases. For example, 88% of the patients with a score >0.65 responded to immunotherapy, while 100% of the patients with a score <0.35 did not respond to immunotherapy. This has been rewritten in the manuscript (**Page 7, Line 17**).

8- The discussion in later parts of the paper appears reliable. The comments that they observe a significant but modest predictive performance of the TNBC – ICI classifier is generally in line with expectations from previous studies predicting response to ICI, in terms of possible AUC, for other cancer types where response has been noted.

Response to Reviewer: Thank you very much for this comment, we appreciated your revision and constructive criticism.

9- Page 10, line 7. Comment: *you have to be careful with the integration of molecular signatures. These may have been derived from data mentioned in the study and could lead to circularity (i.e. an unfair data leakage which could bias the predictor towards over-stated test accuracies because relevant information about the test label is presented to the classifier). This is very unlikely to have happened here since later (I17+, etc), the authors report modest effects when integrating molecular signatures.*

Response to Reviewer: Yes, we agree with your observation and are aware of potential circularity when employing molecular signatures, since they are also derived from data employed in the study. For this reason, we tested each molecular signature independently, in combination with the other molecular signatures, and integrated it with our classifier (**Page 10, Line 12**).

10- Comment: *(not suggesting the authors do this but it is an observation about dataset sizes and possible 'plateau effects', line 10, page 13): all machine learning algorithms follow a Zipf's Law dependence of the empirical test error, z , on the sample size, m , via $z=a m^{-\alpha} +b$ where α is related to the efficiency of the algorithm. If possible, you can train at different m , find z and then use this, and optimization, to find a , b and α . A non-zero b as $m \rightarrow \infty$ can indicate an intrinsic prediction limitation, possibly noise or missing information, but can give insight as to whether much more useable prediction accuracies can be achieved, if more data could be acquired.*

Response to Reviewer: Thank you very much for this explanation. This has not been added to the manuscript due to the scope of the journal but we will consider it and use it in other studies.

11- *Making user-friendly code and a web-interface available would greatly increase support for publishing this paper*.

Response to Reviewer: This is a great comment since for the application of TNBC-ICI to other cohorts, virtually no researchers would employ the full code to recreate the classifier and apply it to their samples. However, at this stage of the work, we believe that further steps, such as additional testing as data is being generated by new clinical trials, are needed before releasing a user-friendly web interface to the "general"

audience. Yet, to increase the usability of the model, a simplified version of the code has been updated in GitHub, including the .RData files so other researchers with minimal R programming knowledge can easily apply the classifier to their samples (**Code Availability Section, Page 25, Line 21**).

The above comments and suggestions are for the purposes of clarity and possible improvements (I'm not suggesting the authors necessarily go with any comments).

Overall, TNBC is a problematic disease with limited current treatments and generally poor outcomes. Hence, given the relative absence of supervised learning methods for ICI response for TNBC, this would be a good supporting case for this paper.

Response to Reviewer: Thank you again for revising the manuscript and for your constructive criticism and education.

Reviewer #3 (Remarks to the Author):

Summary: *The manuscript provides an example of one approach for developing a gene classifier for prediction of clinical outcomes following chemotherapy plus ICI in TNBC.*

Major concerns:

- *As discussed in page 9 starting at line 12, the validation cohort was used to tune the classifier. Therefore, additional validation in an external cohort is necessary. Perhaps it is possible to collaborate with investigators of the neoTRIP or NeoPACT or another trial.*

Response to Reviewer: This is an excellent point. While in our study the cohort of patients employed to train is different from the cohort employed to validate the model, we agree that including additional testing cohorts would benefit the evaluation of the classifier's performance. However, until this moment, and due to the relatively short follow-up of most of the ongoing clinical trials on immunotherapy on early TNBC and data embargo policies, this has not been possible. Yet, in our revised manuscript, we have provided a simplified version of the classifier for any researcher working on clinical studies testing the response of early TNBC to chemotherapy plus immunotherapy to apply, challenge, and improve the TNBC-ICI classifier (**Code Availability Section, Page 25, Line 21**). In addition, while no additional TNBC patients have been tested, the TNBC-ICI classifier shows a modest to good performance in non-TNBC breast cancer (AUC=0.72-0.75, equivalent to other previously published ICI-predictive methods) and even in non-breast cancer tumors treated with immunotherapy plus chemotherapy (median AUC=0.67). Thus, suggesting that this signature involves robust features to predict immunotherapy response.

- *Limited information is provided about how the classifier was compared to other leading gene classifiers, some which are now clinically available (i.e. 27-gene IO score is mentioned briefly in the discussion page 13 line 5.) Please provide a more comprehensive summary of how the comparison was made and include in the results section rather than mentioning it only in the discussion. Also please provide additional summary about other gene classifiers in the literature, including the Agendia Impact classifier developed by ISPY2 investigators (SABCS 2022), and others (for example, the chemokine 12 score presented by Soliman et al at SABCS 2021)*

Response to Reviewer: Thank you for bringing these studies to our attention. We have included these classifiers in the revised manuscript. For example, the comparison with the Iwase's 27-gene IO classifier has been added to the results section (**Page 9, Line 29**). Furthermore, a comparison with the accuracy shown by the subtyping methodology employed by Bagaev et al. has been included in the discussion, and the discussion now includes a section with a comparison with all the mentioned classifiers, among others (Page 13, Line 4). In addition, the signatures based on chemokines expression displayed significant performance in predicting response in ICI-treated TNBC patients, similar to the TNBC-ICI (OR: 2.5-4; **Page 13, Line 20**).

- *Most classifiers are being developed as a prediction biomarker, i.e. to predict benefit of adding ICI to chemo using pCR as the clinical outcome, comparing the chemo v. ICI+chemo arms. The gene classifier in this manuscript is trained as a prognostic biomarker, i.e. to forecast pCR outcome of ICI+chemo. Please justify this approach, and provide more clarity on how then the biomarker would be clinically useful, especially in comparison to the 27-gene IO score which is shown to predict ICI benefit when added to chemo.*

Response to Reviewer: We agree with this observation and believe in the importance of considering different degrees of response to immunotherapy plus chemotherapy for TNBC patients. In this current study, however, we are limited by the data provided by the different clinical studies. As such, the only parameter that was uniformly reported by the surveyed studies is the pCR. As the technique employed to construct the TNBC-ICI requires statistical power, we focused on pCR instead of other measurable responses.

In addition, in the revised version of the manuscript, we assessed whether the TNBC-ICI classifier predicts pCR in TNBC patients treated with chemotherapy alone (n=56). Our results show that this classifier predicts response to immunotherapy plus chemotherapy, and not to neoadjuvant chemotherapy alone (AUC=0.53; **Page 10, Line 30**).

The 27-gene signature was also designed to determine pCR to immunotherapy plus chemotherapy. When comparing the two systems, the TNBC-ICI shows superior performance in determining pCR to immunotherapy plus chemotherapy than the 27-gene signature (AUC=0.89 vs AUC=0.76). This may be due in part to the different methods employed to generate these classifiers. While the 27-gene signature was constructed using differential expression and employing a unique cohort of patients, the TNBC-ICI was built using machine learning on a training cohort and validated on an independent

cohort of TNBC patients. In addition, the code to apply the TNBC-ICI classifier is accessible to scientists to test and improve the performance of this method in new cohorts of TNBC patients (**Code Availability Section, Page 25, Line 21**). This means that the 37 genes included in the model, can gain or reduce importance on the model as the system keeps learning from new data. Therefore, this poses TNBC-ICI as a tool with the potential to improve the clinical management of TNBC patients undergoing immunotherapy plus chemotherapy.

Minor concerns:

- ***Additional grammatical revisions/refinement are necessary before consideration of publication.***

Response to Reviewer: Thank you, grammar and writing style have been revised and corrected in several parts of the manuscript.

- ***Page 4 line 16 – atezo did not meet OS endpoint in overall cohort but there was an OS benefit in PD-L1+ cohort (although this was not a prespecified endpoint per the statistical analysis plan).***

Response to Reviewer: Thank you for pointing this out, this part of the introduction has been corrected to reflect the improved effects in the PD-L1 positive cohort.

- ***Please provide a plausible next step for your investigative plan with the classifier***

Response to Reviewer: The TNBC-ICI classifier is being made publicly available. With this, we aim to facilitate the access of scientists and physicians working on immunotherapy treatment of TNBC that can challenge the classifier. As the time comes, we plan to develop a web-based interface that, following data protection policies, can improve the TNBC-ICI by learning from misclassified outcomes. We, on the other hand, plan to continue improving the classifier by adding additional datasets as these are made publicly available or data embargo policies allow us. In addition to considering pCR as the predictable outcome, we plan to expand the application of this classifier, or a re-trained classification system to predict partial response to chemotherapy plus immunotherapy. We have included these objectives and the next steps in the revised manuscript (**Page 15, Line 1**).

Reviewers' comments:

Reviewer #2 (Remarks to the Author):

The authors have attempted to meet all points made by the reviewer. All reviewers raise the issue of validation of the method on further unseen TNBC data. However, as the authors point out to Reviewer 1 this data is currently absent, and possible alternatives are not really relevant. Of the more substantive issues raised: the authors respond to the issue of code/website availability with a simplified model made available by GitHub (p.25, L21) and have helped clarify descriptions of the machine learning aspects of the report. Taken in the round:

1. this last point of code/data availability limits the prospect of independently verifying the proposed model, but:
2. the performance of the TNBC-ICI predictor is approximately in line with expectations derived from similar ICI predictors evaluated elsewhere, and is in line with biological understanding of TNBC, but:
3. in reality, the success rate for the TNBC-ICI predictor is well below the level where it could be clinically used (more comment on this below), but:
4. this situation has prevailed before, e.g. in the early days of using machine learning/bioinformatics with the first gene expression array datasets, papers presented very limited results and were often naive, but the field rolls forward with more data/better models and the early papers remain often cited, and later models are actually used in clinical practice, also:
5. the paper now includes mention of some other classifiers, such as the IO classifiers, chemokine12, though the present work complements these.

With these accepted limitations the case remains that there is limited literature on predicting immunotherapy response in TNBC patients, responses are observed, and this, along with limited treatment options for TNBC, gives a positive case for the paper.

Two comments:

1. 'clinically used': in comment 7 the reviewer queries if a confidence measure is available. The confidence measure is apparently implemented in the algorithm, and on p. 7 L17 they mention this. This mention is still way too brief and it would appear to be part of a software package. However, this point is important.

Indeed, the proposed tool might conceivably be clinically relevant if restricting to a small subset of patients where sound predictions can be made (the

test accuracy will be higher for high confidence predictions), rather than trying to predict with all patients. This `... while 100% of patients with a score <0.35

did not respond to immunotherapy', would not be relevant if that was 2 patients, but becomes clinically interesting if its 20: how many patients does this apply to?

Also 88% of patients with a score >0.65 responded: how many? This also appears interesting. If implementing a clinically useable predictor you can go for a

three class tool: reliable responder : uncertain : reliable non responder.

2. Given lack of datasets in this context, and the comments above, I think the reviewer 3 comment: `Please provide a plausible next step for your investigative

plan with the classifier', as a valid added discussion.

Reviewer #3 (Remarks to the Author):

- Page 4 line 84- there was a clinically meaningful improvement in OS in PD-L1 cohort in Impassion 130 but the statistical testing was not formally analyzed. Please correct.
- Page 9 –203 the authors still do not address my concern that there is no separate cohort to test the classifier, since they use the validation cohort to refine the gene expression profile and choose the candidate genes. The authors suggest that the code be provided open source for others to validate in their datasets, which I think needs to be clearly emphasized as an important future direction. this is a potent limitation of the work.
- I thank the authors for discussing other gene classifiers such as the 27 gene signature. That signature has been tested and validated in large datasets, whereas the author's classifier has not yet been tested outside of the dataset that was used to construct the classifier, and therefore I would discourage them from using the word "superior"
- Page 12 276- what does it mean "21.3 more likely to achieve a pCR to ICI"

Reviewers' comments

Reviewer #2 (Remarks to the Author):

The authors have attempted to meet all points made by the reviewer. All reviewers raise the issue of validation of the method on further unseen TNBC data. However, as the authors point out to Reviewer 1 this data is currently absent, and possible alternatives are not really relevant. Of the more substantive issues raised: the authors respond to the issue of code/website availability with a simplified model made available by GitHub (p.25, L21) and have helped clarify descriptions of the machine learning aspects of the report. Taken in the round:

1. this last point of code/data availability limits the prospect of independently verifying the proposed model, but:

2. the performance of the TNBC-ICI predictor is approximately in line with expectations derived from similar ICI predictors evaluated elsewhere, and is in line with biological understanding of TNBC, but:

3. in reality, the success rate for the TNBC-ICI predictor is well below the level where it could be clinically used (more comment on this below), but:

4. this situation has prevailed before, e.g. in the early days of using machine learning/bioinformatics with the first gene expression array datasets, papers presented very limited results and were often naive, but the field rolls forward with more data/better models and the early papers remain often cited, and later models are actually used in clinical practice, also:

5. the paper now includes mention of some other classifiers, such as the IO classifiers, chemokine12, though the present work complements these.

With these accepted limitations the case remains that there is limited literature on predicting immunotherapy response in TNBC patients, responses are observed, and this, along with limited treatment options for TNBC, gives a positive case for the paper.

Two comments:

- 1. 'clinically used': in comment 7 the reviewer queries if a confidence measure is available. The confidence measure is apparently implemented in the algorithm, and on p. 7 L17 they mention this. This mention is still way too*

brief and it would appear to be part of a software package. However, this point is important.

Indeed, the proposed tool might conceivably be clinically relevant if restricting to a small subset of patients where sound predictions can be made (the test accuracy will be higher for high confidence predictions), rather than trying to predict with all patients. This `... while 100% of patients with a score <0.35 did not respond to immunotherapy', would not be relevant if that was 2 patients, but becomes clinically interesting if its 20: how many patients does this apply to?

Also 88% of patients with a score >0.65 responded: how many? This also appears interesting. If implementing a clinically useable predictor you can go for a three class tool: reliable responder : uncertain : reliable non responder.

Response to Reviewer: Thanks for the accurate interpretation of the revised manuscript and for pointing out the existing flaws of this project, as well as in other contemporary attempts to improve the predictive models for immunotherapy response in patients with triple-negative breast cancer. We believe that your comments, both in the first revision and in this one, have significantly improved the paper.

Yes, as you mentioned, the *confidence measure*, or, as mentioned in the paper, just *score*, is obtained by the employed R code (both using (a) the *predict* function, parameters: *type=prob*, input: *randomForest.formula* object and the dataframe; and (b) using the *randomForest* function for the training cohort). An additional explanation of the relevance of using a confidence measure has been added to the revised manuscript (**Page 7, Line 17**, and **Page 9, Line 25**).

We have added a new paragraph (**Page 9, Line 25**) where we explain that the classifier is capable of stratifying patients into the three groups that you mentioned, *reliable responder*, *uncertain*, and *reliable non-responder*. Specifically, we have seen that 19 out of 20 patients with a score over 0.65 (*reliable responders*) achieved a pCR, while only one out of nine patients with *reliable non-responders* (n=9, score<0.4) reached pCR. In the *uncertain* group (0.4<score<0.65), we observed mixed outcomes, with eight responders and 13 non-responders. We believe that while this method may not be useful for all patients, it can have relevance for patients with both reliable responders and reliable non-responders scores.

2. Given lack of datasets in this context, and the comments above, I think the reviewer 3 comment: 'Please provide a plausible next step for your investigative plan with the classifier', as a valid added discussion.

Response to Reviewer: In the revised manuscript, we have expanded the discussion to reflect the limitations of the study and to add the next steps of the investigation (**Page 14, Line 10**). We recognize that the lack of a testing dataset is a limitation of this study. We plan to dynamically update TNBC-ICI when new datasets including transcriptomic and properly annotated outcomes data of TNBC patients treated with ICI become available. This will enable us to continue improving the accuracy of our classifier performance. We have also included a contact in the GitHub repository to propitiate and facilitate communication with other researchers to contribute to improving the classifier. In summary, we view the publication of this manuscript as the first step in the development of a robust classifier to predict ICI response in patients with TNBC and we believe that collaborative efforts will be essential.

Reviewer #3 (Remarks to the Author):

- *Page 4 line 84- there was a clinically meaningful improvement in OS in PD-L1 cohort in Impassion 130 but the statistical testing was not formally analyzed. Please correct.*

Response to Reviewer: Thanks for bringing up this mistake. This has been corrected in the revised version of the manuscript (**Page 4, Line 16**).

- *Page 9 –203 the authors still do not address my concern that there is no separate cohort to test the classifier, since they use the validation cohort to refine the gene expression profile and choose the candidate genes.*
- *The authors suggest that the code be provided open source for others to validate in their datasets, which I think needs to be clearly emphasized as an important future direction. this is a potent limitation of the work.*

Response to Reviewer: In the updated discussion section, we have emphasized the importance of validating the study using new datasets, and have taken steps to facilitate the use of TNBC-ICI by other authors (**Page 15, Line 18**).

In this section, we acknowledge that the lack of a testing dataset is a potent limitation of our work. We also explain that the availability of the code in the GitHub repository allows for a dynamic update of the TNBC-ICI as new datasets including reliable transcriptomic and clinical outcomes data of TNBC patients treated with ICI are released. We believe that this approach will encourage collaboration and lead to a more accurate and clinically relevant tool. We thank you for bringing attention to this limitation and hope that our updated discussion section addresses these concerns.

- *I thank the authors for discussing other gene classifiers such as the 27 gene signature. That signature has been tested and validated in large datasets, whereas the author's classifier has not yet been tested outside of the dataset that was used to construct the classifier, and therefore I would discourage them from using the word "superior"*

Response to Reviewer: We have corrected the wording of this phrase in the text (**Page 5, Line 21**, and **Page 14, Line 10**). We agree that the performance of the TNBC-ICI cannot be categorized as *superior* without additional testing steps.

- **Page 12 276- what does it mean “21.3 more likely to achieve a pCR to ICI”**

Response to Reviewer: We apologize for the confusing wording of this phrase. This sentence refers to the odds ratio explained in Figure 3, which indicates that, in our dataset, patients with a high score (upper tertile) had an increased probability of reaching pCR compared to those with a lower score (below upper tertile; **Page 12, Line 12**). We have indicated the limitation of this finding given the lack of additional external testing cohorts. This has also been added to the updated manuscript (**Page 12, Line 31 and Page 15, Line 17**).

REVIEWERS' COMMENTS:

Reviewer #2 (Remarks to the Author):

The authors have answered the queries posed by the reviewer with the current revision.